# TOWARDS ROBUST TRAINING VIA GRADIENT-DIVERSIFIED BACKPROPAGATION

## ABSTRACT

Neural networks are prone to be vulnerable to adversarial attacks and domain shifts. Adversarial-driven methods including adversarial training and adversarial augmentation, have been frequently proposed to improve the model's robustness against adversarial attacks and distribution-shifted samples. Nonetheless, recent research on adversarial attacks has cast a spotlight on the robustness lacuna against attacks targeted at intermediate layers. Towards analyzing the rationale for this robustness lacuna, this paper investigates the layer-wise adversarial effect and adversarial gradients w.r.t intermediate layers. We observe that previous adversarial-driven methods tend to generate limited perturbations in the shallow intermediate layers compared with the deep output layer and there is a domain gap existing between the intermediate layer gradients generated by various adversarial techniques. The observed robustness lacuna can be primarily attributed to the exclusive utilization of loss functions on the output layer for adversarial gradient generation. This inherent practice constrains the adversarial impact on the shallow intermediate layers. Therefore, from the standing point of diversifying the adversarial gradients to ensure sufficient training and robustness of intermediate layers, this paper proposes a novel Stochastic Loss Integration Method (SLIM), which can be instantiated into the existing adversarial-driven methods in a plug-and-play manner. Experimental results across diverse tasks, including classification and segmentation, as well as various areas such as adversarial robustness and domain generalization, validate the effectiveness of our proposed method. Furthermore, we provide an in-depth analysis to offer a comprehensive understanding of layer-wise training involving various loss terms.

## 1 INTRODUCTION

Recent advances in convolutional neural networks (CNN) have enabled remarkable success in various computer vision tasks, including classification, segmentation and object detection He et al. (2016); Mo et al. (2022); Wang et al. (2023). Yet CNNs are prone to be vulnerable against adversarial attacks Goodfellow et al. (2015); Madry et al. (2018) and out-of-distribution (OOD) samples Hendrycks & Dietterich (2019), which constrains the broader application of deep learning. Consequently, extensive studies have been dedicated to improving models' robustness against diverse input perturbations. Various defense strategies, including loss regularization Jeong & Shin (2020); Li & Zhang (2021); Kim et al. (2021), adversarial training (AT) Madry et al. (2018); Zhang et al. (2019a); Wang et al. (2020), and data augmentation Hendrycks et al. (2020; 2022); Huang et al. (2021), have been proposed to defend against adversarial attacks and OOD samples.

Among the aforementioned strategies, adversarial techniques have consistently demonstrated their effectiveness. These techniques encompass adversarial training Madry et al. (2018); Wang et al. (2020) for countering adversarial attacks and adversarial augmentation Wang et al. (2021); Zhang et al. (2023) to improve domain generalization. Adversarial techniques tackle a min-max optimization problem involving the loss function. In this process, they maximize the targeted loss function by introducing perturbed gradients into specific components, such as input images or feature statistics. Subsequently, they minimize this targeted optimization loss function by updating the model's parameters using gradient descent methods, which involve iteratively adjusting model parameters to approach the loss function's minimum. The majority of these methods utilize the cross-entropy (CE) loss or its variants for adversarial sample generation in training. However, recent research in

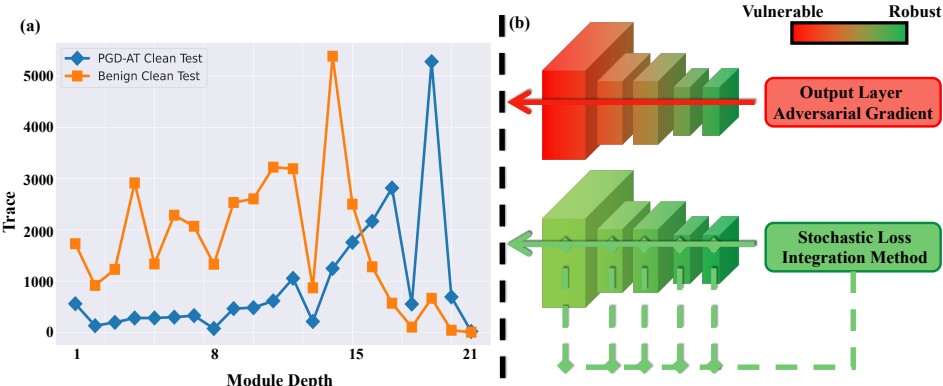

Figure 1: Illustration of (a) vulnerabilities in models trained using traditional adversarial training (PGD-AT Madry et al. (2018)) and (b) our proposed solution towards addressing these vulnerabilities. (a) Module-wise trace of the Hessian matrix Dong et al. (2020) computed as the second-order derivative of the loss function w.r.t. the module parameters. The trace of the Hessian matrix serves as a sensitivity measurement reflecting the module-wise flatness of the loss landscape Keskar et al. (2017); Zhuang et al. (2022). (b) Comparison between adversarial gradients backpropagated from the output layer and adversarial gradients obtained through our SLIM approach.

adversarial attackYu et al. (2021) has uncovered a vulnerability in models trained using the afore-mentioned strategies. These models, while effective against certain adversarial attacks, struggle to maintain robustness when confronted with attacks targeting the intermediate layers of the network Yu et al. (2021); Jin et al. (2023). To uncover the underlying reasons behind this previously undiscovered vulnerability, we conduct a detailed comparison of module-wise traces of the Hessian matrix between a benign model and one that has undergone adversarial training. In most cases, a higher trace of the Hessian matrix corresponds to a flatter loss landscape. A trained model with a smooth loss landscape often exhibits greater robustness and generalization capabilities Dong et al. (2020); Keskar et al. (2017); Zhuang et al. (2022). Based on the analysis presented in Fig. 1(a), it is evident that adversarial training (i.e., PGD-AT) leads to higher traces in only the deep modules of the network compared to those in a benign model. Paradoxically, while these benefits enhance robustness in deeper layers, they simultaneously result in a loss of robustness in the shallower and intermediate layers of the model. This observation could potentially serve as evidence for why adversarial learning ceases to be effective in resisting intermediate layer attacks (ILA).

On the other hand, previous adversarial augmentation methods Wang et al. (2021); Zhang et al. (2023); Zhong et al. (2022) mainly leverage the cross-entropy loss as the adversarial loss term to generate gradients in an augmented feature or image space that be generalized to unseen domains. However, OOD samples may result in shifted distributions within a feature space at various hierarchical levels that may not necessarily approach the output layer. Consequently, previous works have encountered limitations in achieving a comprehensive search space for adversarial augmentation to encompass various types of OOD samples.

In this paper, we undertake a comprehensive analysis of limitations observed in prior adversarial-driven methods that primarily rely on loss functions operating at the output layer. The analysis cover two key aspects: (1) the limited adversarial perturbations in the shallow layers and (2) the salient domain gap between adversarial gradients generated by attacks targeting the output layer and those directed at intermediate layers. We further scrutinize the constrained gradients from the perspective of the chain rule. Subsequently, in the pursuit of ensuring sufficient training for intermediate layers, we introduce an approach aimed at *diversifying the generated adversarial gradients*, termed Stochastic Loss Integration Method (SLIM), as illustrated in Fig. 1(b), which can seamlessly integrate into various adversarial-driven methods in a plug-and-play manner to further boost the performances.

Experimental results show that despite our SLIM is simple and effective, it records state-of-the-art performances when instantiated in adversarial training and adversarial augmentation methods. Investigations of the models' clustering effect Jin et al. (2023) and the trace of the layer-wise Hessian

matrix Dong et al. (2020) provide further insights into how each layer of models is affected in the training process. To summarize, our contributions are three-fold:

- We unveil that the invalidity of adversarial-driven methods against intermediate layer attacks stems from the restricted adversarial gradients resulting from the exclusive use of loss functions at the output layer for adversarial gradient backpropagation.

- We introduce the Stochastic Loss Integration Method (SLIM), which can be seamlessly integrated into existing adversarial-driven methods to further boost the robustness performances. It allows diversifying the generated adversarial gradients for ensuring the robustness of intermediate layers.

- Experimental results demonstrate the effectiveness of SLIM across diverse tasks and research domains. Moreover, we provide interesting insights into layer-wise training under adversarial-driven methods.

## 2 RELATED WORKS

### 2.1 ADVERSARIAL-DRIVEN METHODS

Since the vulnerability of deep learning models against adversarial attacks and OOD samples has been reported Madry et al. (2018); Hendrycks et al. (2022), many works studied the robustness of the models and proposed several defense strategies. Among the various strategies proposed, one of the most effective is known as the adversarial-driven approach. This approach can be further classified into two categories: adversarial training, which focuses on defending against adversarial attacks, and adversarial augmentation, which tackles OOD samples.

**Adversarial Training.** Adversarial training is the most effective way of improving adversarial robustness by adapting adversarial samples for training. Mardy et al.Madry et al. (2018) propose to train the model to minimize the adversarial loss while using PGD attack to maximize it. The vast majority of adversarial training methods follow the same paradigm, but train the model with different loss objective functions to obtain better robustness performances. TRADES Zhang et al. (2019a) minimizes the multi-class calibrated loss between the output of the original image and that of the adversarial examples as the surrogate loss function to substitute cross-entropy loss. MART Wang et al. (2020) proposes to revisit the misclassified samples and optimize the misclassification-aware regularization with the standard adversarial risk.

**Adversarial Augmentation.** Adversarial augmentation has recently been investigated to help models obtain stronger robustness against distribution-shifted samples. It intends to apply data augmentation under the guidance of adversarial gradients instead of randomness. Wang et al. Wang et al. (2021) develop the adversarial variant of AugMix Hendrycks et al. (2020), namely as AugMax to adversarially mix multiple diverse augmented images. Their method achieves a significant improvement in OOD robustness compared to the random mixing of AugMix. Zhang et al. Zhang et al. (2023) formulate AdvStyle to explore a larger augmentation space for feature-level style augmentation with adversarially updating the statics control factors. Similar approaches have also been proposed Zhong et al. (2022); Fu et al. (2023) for cross-domain segmentation and few-shot domain generalization.

### 2.2 ROBUSTNESS LACUNA OF INTERMEDIATE LAYER

Previous adversarial training methods mainly employ loss functions in the output layer solely for adversarial gradient backpropagation Madry et al. (2018) but overlook the importance of intermediate layers. Notably, Feature Scattering Zhang & Wang (2019) generates adversarial examples by maximizing the distances between the adversarial features and the natural ones. Bai et al. Bai et al. (2021) aims to suppress redundant channel activations in intermediate layers by adversarial examples. Recently, LAFEAT Yu et al. (2021), an adversarial attack targeting the intermediate layers, revealed that the intermediate layers features can be effectively utilized for crafting the adversary, indicating the existing robustness lacuna of intermediate layers even for adversarial-trained models. However, we observe that the majority of the adversarial robustness works solely apply output layer attacks during training and evaluation, but do not take intermediate layer robustness into consideration.

# 3 ADVERSARIAL GRADIENTS OF INTERMEDIATE LAYERS

Previous works in adversarial augmentation Wang et al. (2021); Zhang et al. (2023); Zhong et al. (2022) and the majority of works in adversarial training Madry et al. (2018); Zhang et al. (2019a); Wang et al. (2020) primarily employ loss functions at the output layer (e.g. cross-entropy loss) to compute backward adversarial gradients. In this section, we present a comprehensive analysis of the adversarial effect on different layers and the existing domain gap of adversarial gradients generated with output layer losses and intermediate layer losses. The analysis demonstrates that relying solely on output layer losses (OLL) is insufficient for generating substantial perturbations in the intermediate layers during training, resulting in a lacuna of robustness for OOD samples and adversaries.

## 3.1 ADVERSARIAL EFFECT ON DIFFERENT LAYERS

Here, we observe how diverse adversarial attacks or adversarial augmentation techniques impact various layers. To assess the influence of these different adversarial techniques, we measure cosine similarity between the features of benign samples and that of the adversarial samples across different layers to quantify the extent of the adversarial impact.

As shown in Fig. 2, solely adopting OLL to generate adversarial samples may have a limited impact on the shallow layers, indicating that the OLL-generated adversarial samples primarily disrupt features in the deeper layers. By exclusively utilizing these adversarial samples for training, the robustness of intermediate layers cannot be ensured as there is a deficiency in intermediate layer perturbations during the training process. This deficiency can be exploited to create potent intermediate layer perturbations through intermediate layer attacks. As shown in Tab. 1, intermediate layer attack achieves larger performance decline against PGD-trained models compared with PGD-20, validating the existence of robustness lacuna in the intermediate layers for the adversaries. To generate different intermediate perturbations from vanilla PGD (PGD-CE) in the deep semantic space, we introduce a simple variant of PGD (PGD-CS), that employs the cosine similarity loss of the features in the deep semantic layer instead of cross-entropy loss at the

Table 1: Recognition accuracy of adversarially-trained models under output-layer attack (PGD-20 Madry et al. (2018)) and intermediate layers attack (LAFEAT Yu et al. (2021) and SSA Luo et al. (2022)). PGD-CE and PGD-CS denote the PGD training using cross-entropy loss and cosine similarity loss to generate adversarial samples, respectively. PGD-CE and MART Wang et al. (2020) employ output-layer losses for adversarial sample generation and PGD-CS employs an intermediate-layer cosine similarity loss function.

| Model / Attack | PGD-20 | LAFEAT | SSA |
|---|---|---|---|
| PGD-CE AT | 44.66 | 38.07 | 51.21 |
| MART | 48.13 | 41.12 | 49.89 |
| PGD-CS AT | 41.28 | 39.70 | 56.84 |

output layer, to generate adversarial training samples. In this case, PGD-CS aims to minimize the cosine similarity between deep features from adversarial samples and benign samples. As shown in Tab. 1, by employing targeted perturbations on the deep semantic layer, the PGD-CS AT trained model's robustness against intermediate-layer attacks is improved. This indicates that inducing perturbations in the intermediate layers during the training can improve robustness against intermediate-layer attacks.

## 3.2 GRADIENTS OF INTERMEDIATE LAYERS

Since the adversarial effect is determined by the generated adversarial gradients, it is vital to understand the difference between gradients generated by various attack methods. In this section, we investigate the relationship between the OLL-generated gradients and gradients generated with intermediate layer attack (e.g. LAFEAT Yu et al. (2021)). A t-SNE visualization between gradients w.r.t the second layer of ResNet-18 He et al. (2016) generated by PGD and LAFEAT is plotted in Fig. 3. As shown in Fig. 3, a domain gap exists between gradients generated by PGD and LAFEAT, indicating that robustness lacuna

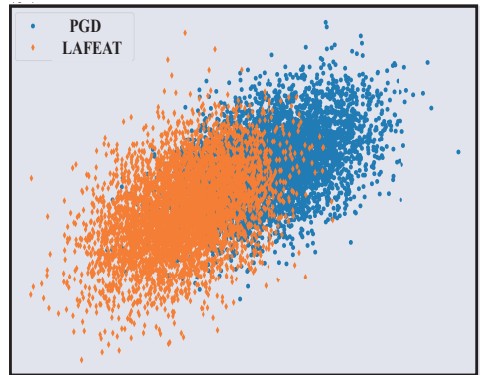

Figure 3: t-SNE visualization of generated adversarial gradients w.r.t the second layer of ResNet-18 by PGD and LAFEAT.

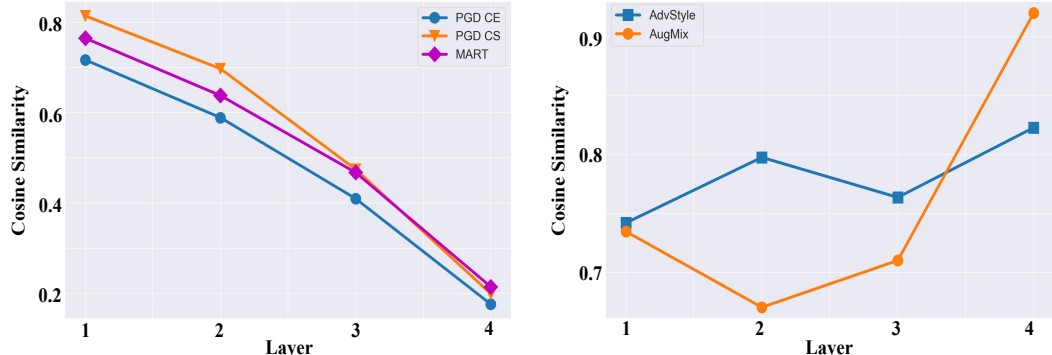

Figure 2: Cosine similarity between adversarial (PGD-CE Madry et al. (2018), PGD-CS and MART Wang et al. (2020)) or data augmentation (AdvStyle Zhang et al. (2023) and AugMix Hendrycks et al. (2020)) features and their benign features across different layers. PGD-CE and PGD-CS denote the PGD attack using cross-entropy loss and cosine similarity loss to generate adversarial samples, respectively. The cosine similarity loss is adopted for the features in the 4-th layer.

existing in the intermediate layers of OLL-driven adversarially trained model can be found by the intermediate-layer attacks. Therefore, to achieve the robustness of intermediate layers, from a data augmentation perspective, more diversified gradients should be generated to equip the model with the ability to resist different attacks targeting various positions.

## 3.3 CONSTRAINT OF THE CHAIN RULE.

As the backward adversarial gradients are computed with the chain rule, in this section, we delve deeper into our analysis by considering the chain rule perspective. Eq. 1 describes the intermediate layer features adversarial gradient backpropagation solely driven by an output layer loss, where $g^l$, $\mathcal{L}_{\text{OLL}}$ and $f_\theta^l(x)$ denote the gradient of the features from the $l$-th module, loss functions applied at the output layer and the features in the $l$-th intermediate layer, respectively. As shown in Eq. 1, backward adversarial gradients of the $l$-th layer features are largely determined by the gradients of the latter features from the $l + 1$-th layer.

$$g^l = \frac{\partial \mathcal{L}_{\text{OLL}}(f_\theta^L(x), y)}{\partial f_\theta^l(x)} = g^{l+1} \cdot \frac{\partial f_\theta^{l+1}(x)}{\partial f_\theta^l(x)} \tag{1}$$

In this case, the adversarial backward propagation is completely driven by the output-layer loss (e.g. cross-entropy loss). Since the gradients take the steepest direction to solely maximize the output layer loss for adversarial sample generation regardless of the adversarial effect in the intermediate layers, the searching space of the adversarial gradients would be severely constrained. From the standing points of improving the robustness of intermediate layers, it would be intuitive to consider adopting an additional loss term explicitly functioned on the features of intermediate layers for inducing more sufficient intermediate layer perturbations in the training.

## 4 STOCHASTIC LOSS INTEGRATION METHOD (SLIM)

With solely leveraging OLL to generate adversarial gradients having a limited effect in the intermediate layers, we propose to introduce an additional and random loss term functioned in the intermediate layers to combine with OLL to calculate adversarial gradients, namely as Stochastic Loss Integration Method (SLIM). It is worth noting that the proposed method serves as a plug-and-play strategy that can be easily instantiated into methods in adversarial training and adversarial augmentation. However, for the sake of clarity, we illustrate how our method is inserted into adversarial training. To provide a sufficient search in the adversarial gradient generation space, the proposed SLIM mainly has two random elements: arbitrary functioned layer and stochastic loss function metric.

**Arbitrary layers for inducing perturbation.** Let $f_\theta^L$ be a neural network with $L - 1$ intermediate layers and parameter $\theta$. $x^{(l)}$ and $f_\theta^{(l)}(x)$ denote the input and output of the $l$-th intermediate layer.

It is worth noting that all layers, excluding the final layer which outputs the probability vectors, are referred to as intermediate layers in the former definition.

To ensure a comprehensive exploration of perturbations across various intermediate layers during the training process, we randomly assign the $l$-th intermediate layer as the position to adversarially induce perturbation.

**Diversity of intermediate layer perturbations**. In order to empower the model with the ability to withstand a broad range of potential disruptions, the additional loss term functioned in the chosen intermediate layer is randomly sampled from a formulated dictionary $D_{d_n}$ with $d_n$ various loss function metrics, including widely used mean square error loss and cosine similarity.

Each time we generate adversarial samples, as shown in Eq. 2, we employ both the task-specific OLL $\mathcal{L}_{\text{task}}$(e.g. cross-entropy loss) and the selected intermediate layer loss function $\mathcal{L}_{\text{dict}}$ for the computation of adversarial backward gradients. To ensure a sufficient exploration of the adversarial augmentation space, we introduce an additional mixing factor $\lambda \sim U(-1, 1)$ to control the respective influences of the two loss components in generating adversarial gradients.

$$\mathcal{L}_{adv} = \mathcal{L}_{\text{task}}(f_\theta^L(x), y) + \lambda \cdot \mathcal{L}_{\text{dict}}(f_\theta^l(x), f_\theta^l(x^{adv})) \tag{2}$$

In the context of adversarial training, as shown in Eq.3, we iteratively update the input image $x$ utilizing the adversarial gradients generated by the adversarial loss combination defined in Eq.2 for $t_{adv}$ times.

$$x_{t+1}^{adv} = x_t^{adv} + \alpha \cdot sign(\nabla_{x_t^{adv}} \mathcal{L}_{adv}(x_t^{adv}, y; f)) \tag{3}$$

To incorporate the proposed method into adversarial augmentation or domain adversarial training, one will only need to adopt the adversarial loss combination to update the corresponding elements (e.g. the control factor of feature statics in adversarial feature-level style augmentation for domain generalization Zhang et al. (2023)) instead of the input images for adversarial training.

## 5 EXPERIMENTS

**Experiment Setup.** To validate the effectiveness of the proposed method, we integrate it into diverse tasks across two research domains: (i) cross-domain classification and segmentation for domain generalization and (ii) adversarial robustness evaluation for adversarial training.

**Implementation Details.** For domain generalization, reported results in both multi-source and single-source domain generalization are averaged over three runs. We train models for 120 epochs across all datasets. For adversarial training, we set PGD as $\epsilon = 8/255$ and $\alpha = 1/255$ for PGD-AT Madry et al. (2018). When integrating SLIM with other methods Wang et al. (2020); Zhang et al. (2019a); Zhong et al. (2022), we follow the same hyperparameter settings of the original literature. More implementation details can be found in the appendix.

### 5.1 DOMAIN GENERALIZATION

In the area of domain generalization, we integrate the proposed method into AdvStyle Zhang et al. (2023), which is a feature-level adversarial style augmentation method for domain generalization. To solidify the effectiveness of the proposed method, we conduct experiments on the task of domain generalized classification and segmentation. We evaluate the proposed method under both the leave-one-domain-out scenario and the more challenging single-source domain generalization scenario.

For multi-source domain generalization, under the widely adopted leave-one-domain-out setting, experiment results are shown in Tab. 5.1. By integrating with SLIM, AdvStyle can be further boosted across four different datasets and two network architectures. Specifically, when conducting experiments with ResNet-18 He et al. (2016), performances of AdvStyle on the PACS and OfficeHome dataset can be further improved by the margins of 3.12% and 2.24%, respectively. For the more challenging single-source domain generalization scenario, experiment results are shown in Tab. 5.1. When integrated with the proposed SLIM, the performances of AdvStyle can still be boosted across

various datasets in the two tasks of segmentation and classification, which further validates the effectiveness and versatility of the proposed SLIM.

| Method | PACS | VLCS | OfficeHome | TerraIncognita |
|---|---|---|---|---|
| **ResNet-18** | | | | |
| Baseline | 79.68 | - | - | - |
| FACT$_{CVPR'21}$ | 84.51 | - | 66.56 | - |
| StyleNeophile$_{CVPR'22}$ | 85.47 | - | 65.89 | - |
| COMEN$_{CVPR'22}$ | 85.70 | 75.00 | 66.50 | - |
| MVDG$_{ECCV'22}$ | **86.56** | **77.13** | 66.80 | - |
| DSU$_{ICLR'22}$ | 82.70 | - | 66.10 | - |
| AdvStyle$_{arXiv'23, NeurIPS'22}$ | 83.00 | 74.86 | 66.48 | 43.32 |
| AdvStyle + SLIM | 86.12 | 76.03 | **68.72** | **45.95** |
| **ResNet-50** | | | | |
| DAC-P$_{CVPR'23}$ | 85.60 | 77.00 | 69.50 | 45.80 |
| AdvStyle$_{arXiv'23, NeurIPS'22}$ | 84.72 | 75.89 | 67.94 | 44.31 |
| AdvStyle + SLIM | **87.03** | **77.53** | **69.21** | **46.05** |

Table 2: Experiment results of multi-source domain generalization on classification under the leave-one-domain-out setting on PACS Li et al. (2017), VLCS Fang et al. (2013), OfficeHome Venkateswara et al. (2017) and TerraIncognita Beery et al. (2018).

| Method | Segmentation (mIoU) | Classification (Acc.) | |
|---|---|---|---|
| | GTA5 → Cityscapes | PACS | CIFAR-10-C |
| Baseline | 37.0 | 46.6 | 74.2 |
| pAdaIN | 38.7 | 51.7 | 76.4 |
| MixStyle | 38.8 | 51.7 | 76.6 |
| DSU | 40.3 | 53.7 | 76.6 |
| AdvStyle | 41.9 | 58.7 | 78.0 |
| AdvStyle + SLIM | **44.1** | **67.1** | **80.6** |

Table 3: Experiment results of single-source domain generalization on classification and semantic segmentation. ResNet-101 (Deeplab v2), ResNet-18 and WideResNet-40-2 are adopted as the baseline setting for segmentation in GTA5 Richter et al. (2016) to Cityscapes Cordts et al. (2016) and classification in PACS Li et al. (2017) and CIFAR-10-C Hendrycks & Dietterich (2019), respectively.

## 5.2 ADVERSARIAL TRAINING

In this section, we integrate the proposed method into widely used adversarial training methods Madry et al. (2018); Zhang et al. (2019a); Wang et al. (2020) and evaluate the adversarial robustness against both output-layer attacks (PGD Madry et al. (2018), AutoAttack Croce & Hein (2020)) and intermediate-layer attacks (LAFEAT Yu et al. (2021), SSA Luo et al. (2022)).

Tab. 4 describes the comparison of the adversarial robustness of the neural networks trained by various methods on the CIFAR-10 dataset Krizhevsky et al. (2009). As shown in Tab. 4, the performances of adversarial training methods can be further boosted by integrating with the proposed SLIM in terms of robustness against both output layer attacks and intermediate layer attacks. Specifically, the robustness of MART has been improved by 1.15% and 3.34% in terms of robustness against PGD-20 and LAFEAT, respectively. The experiment results of robustness against intermediate layer attacks further demonstrate that the proposed method can ensure the strong robustness of intermediate layers.

| Method | Clean | PGD-20 | AutoAttack | LAFEAT | SSA |
|---|---|---|---|---|---|
| ResNet-18 | 90.70 | 0.00 | 0.00 | 0.00 | 31.79 |
| PGD-AT | 75.82 | 44.66 | 38.07 | 38.68 | 51.21 |
| TRADES | 78.62 | 48.59 | 45.34 | 40.08 | 50.78 |
| MART | 77.87 | 48.13 | 41.76 | 41.12 | 49.89 |
| PGD-AT + SLIM | 79.84 | **51.46** | 38.71 | 43.67 | 63.76 |
| TRADES + SLIM | 79.34 | 49.88 | **45.86** | 43.91 | **68.73** |
| MART + SLIM | **80.26** | 49.28 | 42.05 | **44.46** | 57.21 |

Table 4: Adversarial robustness evaluation on CIFAR-10 Krizhevsky et al. (2009) when the proposed SLIM is integrated with widely adopted adversarial training methods, including PGD-AT Madry et al. (2018), TRADES Zhang et al. (2019a) and MART Wang et al. (2020).

## 6 ANALYSIS

In this section, we conduct an analysis of (1) the models' clustering effect Jin et al. (2023), (2) the average trace of the module-wise Hessian matrix Dong et al. (2020) and (3) the model-wise convergence minimum Zhang et al. (2019b). The clustering effect provides a measurement of the model's layer-wise resistance ability against perturbations. Meanwhile, the average trace of the module-wise Hessian matrix and the model-wise parameter convergence minimum provide insights into the module-wise and model-wise smoothness of the loss landscape, respectively.

### 6.1 CLUSTERING EFFECT

Previous work Jin et al. (2023) introduces the clustering effect as a measurement of models' class-wise resistance ability against noises. In this section, we provide analysis from the perspective of enhancing the clustering effect to improve model robustness.

In this section, we compare the clustering effect accuracy of models trained with different methods under clean test set, adversarial attacks and OOD samples. Experiment results are shown in Tab. 5. Intuitively, a stronger clustering effect accuracy indicates the strong robustness of intermediate layers in resisting diverse perturbations. As shown in Tab. 5, models trained with the proposed method manifest a stronger clustering effect. Specifically, when undertaking intermediate-layer attacks of LAFEAT, clustering accuracy in the 3rd layer of our model significantly outperforms vanilla PGD-AT with a margin of 8.58%. With a stronger clustering effect achieved by models trained with different adversarial-driven strategies when in-

| Method | Layer 1 | Layer 2 | Layer 3 | Layer 4 |
|---|---|---|---|---|
| Clean Test Set | | | | |
| PGD-AT | 32.24 | 39.71 | 57.02 | 77.82 |
| PGD-AT + SLIM | 34.19 | 45.19 | 61.98 | 77.94 |
| AdvStlye | 45.61 | 57.02 | 73.98 | 88.04 |
| AdvStyle + SLIM | **48.28** | **62.32** | **80.18** | **92.16** |
| PGD Adversarial Test | | | | |
| PGD-AT | 28.42 | 32.31 | 35.66 | 22.26 |
| PGD-AT + SLIM | **29.28** | **35.31** | **40.49** | **37.77** |
| LAFEAT Adversarial Test | | | | |
| PGD-AT | 26.33 | 29.70 | 32.52 | 28.17 |
| PGD-AT + SLIM | **28.74** | **32.57** | **41.10** | **37.19** |
| OOD Corruptions Test | | | | |
| AdvStyle | 28.47 | 32.05 | 47.63 | 56.91 |
| AdvStyle + SLIM | **30.98** | **36.65** | **51.38** | **67.69** |

Table 5: Clustering accuracy of models under various test samples.

tegrated with the proposed method, we can demonstrate the effectiveness and universal applicability of the proposed method in improving intermediate layer robustness.

### 6.2 AVERAGE TRACE OF MODULE-WISE HESSIAN MATRIX

In this section, we investigate the average trace of the module-wise Hessian matrix of the model's parameter, which is the second-order derivative of the loss function w.r.t the model's parameters. The average trace of the Hessian matrix can provide insights into the local geometry of the loss landscape

as a sensitivity metric Dong et al. (2020). It is widely acknowledged that a higher trace of the model indicates a smoother local loss landscape, representing stronger robustness and generalization ability Keskar et al. (2017); Zhuang et al. (2022). Clean test samples and adversarial samples are leveraged for calculation separately. Experiment results are shown in Fig. 4. As shown in Fig. 4, (i) Compared with adversarial training, naive training protocol can obtain higher trace in the shallow layers than the adversarial-training methods but suffers a significant drop in the deep layer. (ii) In both testing scenarios, higher traces of the Hessian matrix for the intermediate layers are obtained by PGA AT with SLIM compared with vanilla PGD AT, indicating the generated perturbations targeting the intermediate layers can indeed help the model's intermediate layers converging to a relatively smooth loss landscape. (iii) In the deep layer, the vanilla PGD AT model obtains higher traces than the others, indicating that the vanilla adversarial training method mainly impacts the robustness of the deep semantic layer.

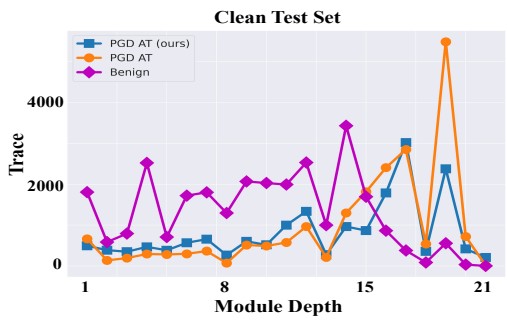 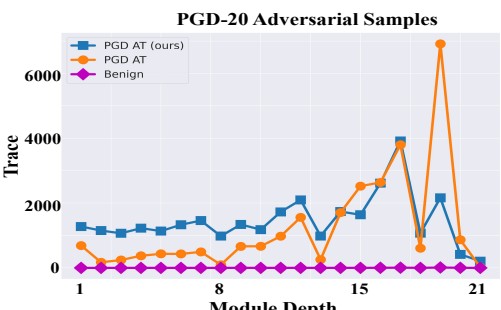

Figure 4: Trace of module-wise Hessian matrix comparison when testing with clean test set and PGD-20 adversarial samples.

## 6.3 FLATTER MINIMUM

Following Zhang et al. Zhang et al. (2019b), we conduct the experiments of injecting Gaussian noise into the trained model weights to show that integrating with the proposed method can lead the convergence of the model to a flatter minimum. As shown in Fig. 5, Gaussian noise is injected into the models' parameters and their test accuracy is plotted. Since the PGD AT model integrated with the proposed method can resist stronger perturbations in the parameters before collapsing, we can conclude that PGD AT model integrated with SLIM is flatter. Since it's universally acknowledged that a flatter minimum guarantees stronger robustness and generalization ability, models trained

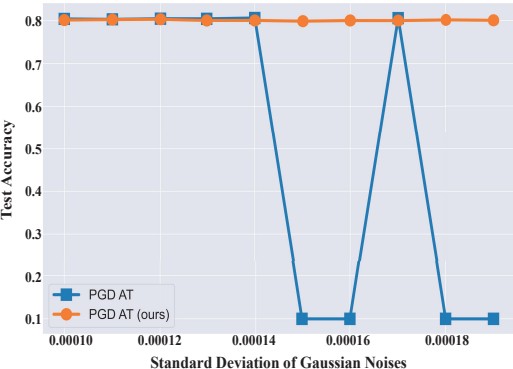

Figure 5: Comparison of test accuracy with increasing Gaussian noise injected into the models' parameters.

with adversarial-driven strategies integrated with the proposed method can obtain stronger robustness and generalization ability.

## 7 CONCLUSION

In this paper, we provide an analysis of the robustness lacuna of intermediate layers from the perspective of adversarial effect, domain gaps of gradients generated with different methods and the chain rule of backward propagation. Thereafter, to ensure the robustness of intermediate layers, we propose SLIM, the stochastic loss integration method, which can integrate into previous adversarial-existing methods of adversarial robustness and domain generalization to further boost the performances. Experiment results of domain generalization and adversarial training demonstrate the effectiveness and versatility of the proposed SLIM. We further provide insights into layer-wise adversarial training.

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

## A    APPENDIX

### A.1    IMPLEMENTATION DETAILS

For domain generalization, in both the multi-source leave-one-domain-out and single-source domain generalization scenarios, we train with the Adam optimizer. In classification tasks on PACS Li et al. (2017), VLCS Fang et al. (2013), OfficeHome Venkateswara et al. (2017), CIFAR-10-C Hendrycks & Dietterich (2019) and TerraIncognita Beery et al. (2018), the initial learning rate and batch size are set as 2e-4 and 64, respectively. A cosine annealing schedule for adjusting the learning rate is also applied. For the cross-domain segmentation task, we follow the same hyperparameter setting with AdvStyle Zhang et al. (2023).

For adversarial training, we follow the attack setting and learning rate used in previous methods Wang et al. (2020); Zhang et al. (2019a). The batch size is set as 64 across all the adversarial training experiments and no data augmentation techniques are included.

### A.2    LAYER-WISE ADVERSARIAL EFFECT COMPARISON

In this section, we investigate the adversarial perturbations induced in the intermediate layers with vanilla PGD Madry et al. (2018) and PGD integrated with the proposed SLIM. As shown in Fig. 7, PGD integrated with SLIM can have a stronger effect in the intermediate layers compared with vanilla PGD.

### A.3    OVERFITTING TO THE ADVERSARIAL OUTPUT LAYER LOSS

Jin et al. Jin et al. (2023) conduct the experiments of calculating the cosine similarity between the intermediate-layer gradient versus the shift caused by the generated perturbation to demonstrate that the adversarial gradients are overfitted to the applied output layer loss.

Following Jin et al. Jin et al. (2023), we conduct the same experiments in Fig. 7. As shown in Fig. 7, by integrating with the proposed SLIM, numerical oscillation of cosine similarity between features shift and the adversarial gradients will occur later compared with the vanilla PGD and TRADES. This indicates that the proposed SLIM can prevent the adversarial gradients from overfitting to the applied loss combinations.

**Shallow layer**                                                    **Deep layer**

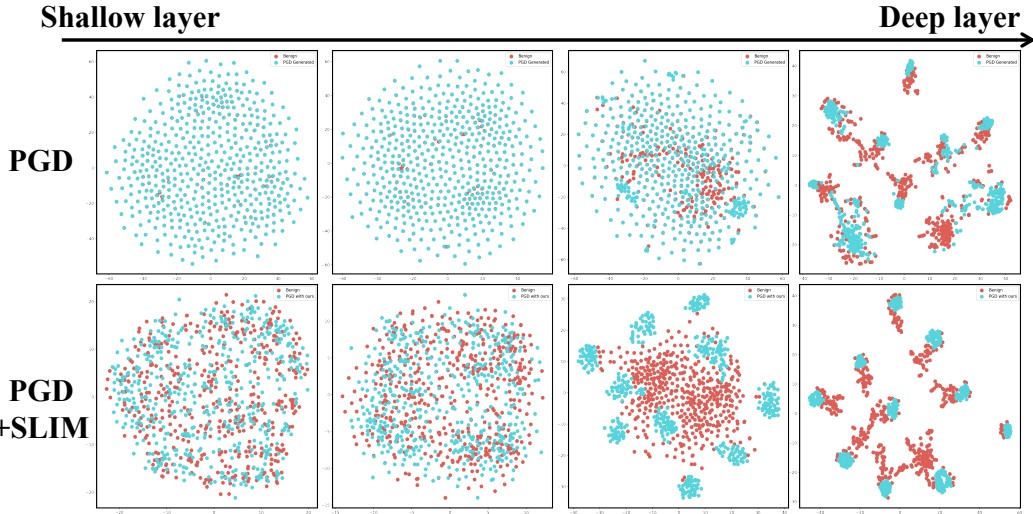

Figure 6: t-SNE visualizations of layer-wise features of PGD and PGD integrated with SLIM from the same class of CIFAR-10 Krizhevsky et al. (2009). Red and blue dots denote the features from benign samples and adversarial samples, respectively.

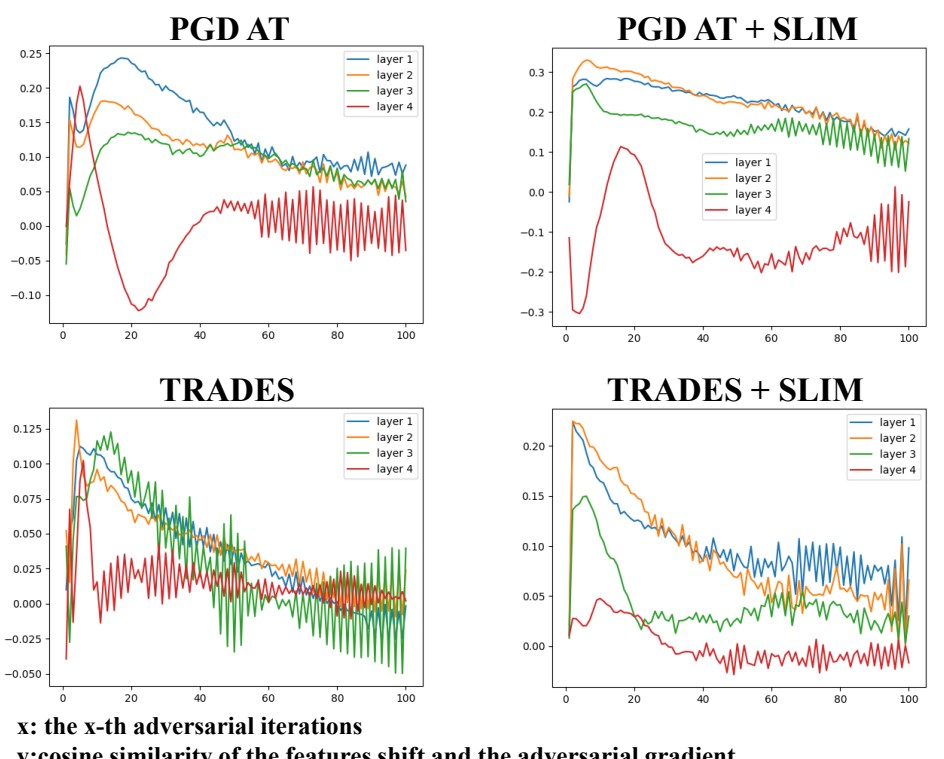

Figure 7: Layer-wise cosine similarity between the intermediate layer adversarial gradient and the feature shift caused by the generated perturbations. Experiments are conducted on the CIFAR-10 Krizhevsky et al. (2009) with ResNet-18 He et al. (2016).

## A.4   SCALABILITY OF LOSS DICT $D_{d_n}$

In this section, we provide an ablation study on the formulated loss functions included in the loss dictionary $D_{d_n}$. For the sake of clarity, only three loss functions are included for all the experiments conducted in this paper, including MSE loss, orthogonal loss and reverse loss as shown in Eq. 4.

$$\begin{cases} \mathcal{L}_{\text{MSE}} = \|f_\theta^l(x) - f_\theta^l(x^{adv})\|_2^2 \\ \mathcal{L}_{\text{Ortho}} = \|\dfrac{f_\theta^l(x) \cdot f_\theta^l(x^{adv})}{\|f_\theta^l(x)\|\|f_\theta^l(x^{adv})\|}\|_2^2 \\ \mathcal{L}_{\text{Reverse}} = -(\dfrac{f_\theta^l(x) \cdot f_\theta^l(x^{adv})}{\|f_\theta^l(x)\|\|f_\theta^l(x^{adv})\|}) \end{cases} \quad (4)$$

Table 6: Ablation study of the formulated loss functions in the loss dictionary $D_{d_n}$ on the PACS dataset for single-source domain generalized classification.

| Loss Dictionary Setting | PACS |
|---|---|
| $D_{d_n}$ | 86.12 |
| $D_{d_n}$ wo $\mathcal{L}_{\text{MSE}}$ | 85.46 |
| $D_{d_n}$ wo $\mathcal{L}_{\text{Ortho}}$ | 85.39 |
| $D_{d_n}$ wo $\mathcal{L}_{\text{Reverse}}$ | 85.02 |

Since a larger loss dictionary can bring more diversified perturbations, we conduct an ablation study on the single-source domain generalization of classification on PACS to validate the high scalability of the proposed method. As shown in Tab. 6, performances on the cross-domain classification integrated with AdvStyle Zhang et al. (2023) decline when removing pre-defined loss functions in the loss dictionary, indicating that the diversity of the generated perturbations in the intermediate layers can help improve the layer-wise robustness and generalization ability. Considering the extension-friendly features of the loss dictionary, with more loss functions appended, our method may boost the performances of the existing adversarial-driven methods even further.

