# OpenReview forum: "Towards Robust Training via Gradient-diversified Backpropagation"
_ICLR.cc/2024/Conference — ICLR 2024 Conference Withdrawn Submission_

### Official Review · Reviewer_Ss1r · 2023-10-28

**Soundness:** 3 good
**Presentation:** 2 fair
**Contribution:** 2 fair
**Rating:** 3
**Confidence:** 4

**Summary:**

This work investigates the robustness of the intermediate layer in adversarial training and attempts to enhance the network's adversarial robustness by strengthening the robustness of the intermediate layer.

**Strengths:**

1. The research problem is intriguing; the robustness achieved through adversarial training varies across different layers of the network, which is an interesting phenomenon. The authors also provide some experimental evidence to support this observation.
2. The experiments are extensive, encompassing results from two domains: domain generalization and adversarial robustness.

**Weaknesses:**

1. Puzzling analysis: In Figure 1, the authors analyze that a higher trace of the Hessian matrix corresponds to a flatter loss landscape, and a model with a smooth loss landscape exhibits greater robustness and generalization capabilities. So, why is the trace of the benign model significantly higher than that of the robust model?
2. Over-claimed Contributions：The authors claim that adversarial-driven methods against intermediate layer attacks are invalid, which is inaccurate. Adversarial-driven methods may simply have relatively weaker robustness at the intermediate layer. Furthermore, the authors claim that the proposed SLIM method achieves state-of-the-art performance. However, the adversarial robustness experimental results provided by the authors are unconvincing and far from state-of-the-art.
3. Lack of Motivation: The weaker robustness of the network's shallow layers can be attributed to their proximity to the input, resulting in less interference from adversarial noise at these layers. The overall robustness of the network may only require relatively weaker robustness at the shallow layers. So, why would enhancing the robustness of the shallow layers improve the overall network robustness? What is the motivation? This is the question that the authors should address in the paper, rather than conducting extensive analysis on the existing phenomenon of weaker robustness in the network's shallow layers.
4. In the methodology, the intermediate layer loss function should be explicitly formulated in the main paper.
5. Confusing Experimental Setup: The experimental setup for the PGD-AT baseline used by the authors is different from the classical setup, including step size, data augmentation, and epoch. It is strongly recommended that the authors report experimental results using the standard PGD-AT baseline, and simultaneously report both the best accuracy and last accuracy.
6. Marginal Robustness Improvement: The improvement in adversarial robustness achieved by the proposed method is very marginal. Across three AT baselines, the improvement from AA is only 0.64, 0.52, and 0.29 respectively. Moreover, the experimental setup is confused.

**Questions:**

Please refer to the questions in the Weaknesses.

---

### Official Review · Reviewer_Rp2G · 2023-10-31

**Soundness:** 2 fair
**Presentation:** 2 fair
**Contribution:** 2 fair
**Rating:** 3
**Confidence:** 4

**Summary:**

The paper shows that with adversarial perturbations computed from outer layer losses(OLL) such as cross-entropy loss, the maximal perturbations occur at the deeper layers, but the perturbations at shallower layers are very small. This is shown by computing cosine similarity between clean image features and adversarial image features at different layers. The cosine similarity is small at deeper layers but larger at the shallower layers. This is the reason, authors argue that intermediate feature adversarial attacks, such as LAFEAT, are effective even against adversarial trained models.
The authors thus propose an adversarial training method, where the adversarial perturbations are also generated to minimize the cosine similarity at shallower layers.

With the proposed methodology the authors show improvement in adversarial robustness and also improvement in domain generalization. Through further ablation the authors further show that with their proposed method, the models exhibit a higher trace of module-wise Hessian matrix for shallower layers thus indicating a flatter minima, which in turn increases the model's robustness. The models are further shown to be robust to random noises in model parameters, further confirming a flatter minima.

**Strengths:**

1. I like the ablation showing robustness to Gaussian noise in model parameters to indicate flatter minima and the module-wise traces of the Hessian matrix at different layers to again denote a flatter loss curve. Its interesting to see this correlation between a flat minima and increased robustness.

**Weaknesses:**

1. The paper's idea of improving robustness in different intermediate layers of a neural network to improve the model's robustness has already been explored in other papers such as [1]. In [1] the authors show a similar conclusion, that the shallower layers of a network are vulnerable to adversarial perturbation and explicitly improving the robustness of these intermediate layers improves the model's robustness and also it's clean accuracy. The paper fails to cite [1] and thus the novelty of this paper is thus limited.

2. Some of the terminologies introduced in the paper are difficult to understand as not much has been mentioned about them. For instance, it was quite hard to understand what is PGD-CE and what is PGD-CS. The authors should either have a better naming convention or put more emphasis into explaining these terminologies.

3. Methods such as [2] explicitly try to make the loss curve flatter. Do these methods automatically lead to improved robustness? How does the proposed idea(SLIM) work with such methods?

References -

1. Harnessing the Vulnerability of Latent Layers in Adversarially Trained Models . Kumari et al. https://arxiv.org/pdf/1905.05186.pdf

2. Sharpness-Aware Minimization for Efficiently Improving Generalization. Foret et al. https://arxiv.org/abs/2010.01412

**Questions:**

I have already mentioned it in the weakness section

---

### Official Review · Reviewer_hkrG · 2023-11-01

**Soundness:** 2 fair
**Presentation:** 3 good
**Contribution:** 2 fair
**Rating:** 6
**Confidence:** 2

**Summary:**

this paper delves into the vulnerabilities of neural network models, especially when faced with adversarial attacks targeting intermediate layers. The authors propose the Stochastic Loss Integration Method (SLIM) as a novel solution to diversify adversarial gradients and enhance the robustness of these layers. Experimental evidence supports the effectiveness of SLIM across various tasks and domains.

**Strengths:**

**Originality & Significance**: The paper introduces the innovative Stochastic Loss Integration Method (SLIM) that addresses vulnerabilities in the intermediate layers of neural networks.

**Quality**: Through a comprehensive analysis and empirical evaluation across diverse tasks and areas, the paper ensures a thorough validation of the SLIM method's effectiveness.

**Clarity**: The paper is well-structured and easy to follow.

**Weaknesses:**

**Limited Comparisons with Existing Methods**: While the paper introduces SLIM and showcases its effectiveness, it would be beneficial to see more detailed comparisons with other state-of-the-art adversarial techniques such as AWP [1], WOT[2]

**Lack of comprehensive validation**:  the evaluation of SLIM+Adversarial training methods are conducted on one simple dataset CIFAR-10 and one small model ResNet-18. It is unknown how does its performance on complex datasets such as Tiny-ImageNet and large architecture such as WideResNet.

[1] Wu, Dongxian, Shu-Tao Xia, and Yisen Wang. "Adversarial weight perturbation helps robust generalization." Advances in Neural Information Processing Systems 33 (2020): 2958-2969.

[2] Huang, Tianjin, et al. "Enhancing Adversarial Training via Reweighting Optimization Trajectory." Joint European Conference on Machine Learning and Knowledge Discovery in Databases. Cham: Springer Nature Switzerland, 2023.

**Questions:**

(1) Can the superior performance of SLIM on adversarial training variants be generalized to complex tasks such as tiny-imagenet and other architectures?

(2) why the AA accuracy reported in this paper is much lower than the AA accuracy reported in [1].

[1] Wu, Dongxian, Shu-Tao Xia, and Yisen Wang. "Adversarial weight perturbation helps robust generalization." Advances in Neural Information Processing Systems 33 (2020): 2958-2969.

---

### Official Review · Reviewer_B9kH · 2023-11-01

**Soundness:** 2 fair
**Presentation:** 3 good
**Contribution:** 2 fair
**Rating:** 3
**Confidence:** 4

**Summary:**

This paper presents a new algorithm for adversarial training and adversarial augmentation to address the problems of adversarial robustness and domain generalization. Specifically, the authors are concerned about the robustness against attacks targeting intermediate layers, such as LAFEAT. The novelty of this work lies in the introduction of random intermediate layer perturbations for defense.

**Strengths:**

The paper is well written and easy to understand.
The focus on intermediate layer attack is interesting and important.

**Weaknesses:**

1. The relationship between the trace of the Hessian matrix and the smoothness of the loss landscape in Fig.1 (a) and Section 6.2 needs clarification. The authors state that "It is widely acknowledged that a higher trace of the model indicates a smoother local loss landscape, representing stronger robustness and generalization ability." However, this statement appears to be fundamentally flawed. Numerous studies, including [1], [2], [3], and Zhuang et al. (2022), unanimously argue that a higher dominant eigenvalue of the Hessian matrix actually compromises network robustness. Furthermore, the experiment results in Fig.1 (a) and Fig. 4, which seem to support the authors' claim, need more explanations as it is widely acknowledged that computing the exact Hessian matrix is very expensive [2]. The following problems should be addressed: is the Hessian computed with respect to the network parameters and how to compute the exact Hessian when there are millions of parameters in ResNet-18? Alternatively, please provide the specific algorithm if the approximated trace of the Hessian is needed.

2. Since the authors do not provide the backbone in the implementation details, I assume the backbone is ResNet-18 based on the context. However, both the natural accuracy and adversarial accuracy under PGD-20 are much lower than those reported in another paper [4]. Clean accuracy is 75.82 compared to 82.7, and PGD-20 accuracy is 44.66 compared to 47.4. Both [4] and this paper use ResNet-18 as the backbone, so I suggest that the authors provide a detailed explanation for the significant performance gap.

3. Regarding the algorithm SLIM provided in Section 4 and Appendix A.4, I have some questions that need to be addressed:
- What is the purpose of inducing perturbations on a random layer? Why not inject perturbations on all intermediate layers or focus more on shallow layers?
- The mixing parameter \lambda is drawn from a uniform distribution U(-1,1). Why does the distribution have a negative part? If the goal is to minimize the cosine similarity when constructing adversarial samples, the positive (or negative, see question below) factor before the second loss is detrimental. If this specific choice of \lambda indeed benefits the experimental results, the authors should provide an ablation study on \lambda and discuss the rationale behind the choice.
- Is it necessary to randomly perturb x before starting the iteration in Eq. (3)? It seems that x is a global maximizer with zero gradient to the objective function (cosine similarity).
- In Appendix A.4, what is the intuition behind the orthogonal loss and reverse loss? The orthogonal loss seems to be the norm of a scalar; please double-check. Additionally, if there is indeed a minus sign before the reverse loss, the authors should explain the negative part of the uniform distribution U(-1,1) from which \lambda is drawn.

4. I would like to see more discussion about applying SLIM in the domain generalization problem. How is SLIM integrated with adversarial augmentation methods like AdvStyle? Why is SLIM useful for domain generalization? The connection between intermediate layer perturbations and domain generalization performance seems unclear.

5. Typos
- "attackYu" in the first line of page 2. A space is needed.
- Double reference when citing papers, e.g., "Wang et al Wang et al" in the Related works - Adversarial Augmentation section. Similar errors appear multiple times throughout the paper.

6. Figures:
- Fig. 3 appears on page 4, while Fig. 2 appears on page 5. I recommend adjusting the order according to the first appearance, or placing Fig. 2 at the top of page 4.
- In Fig. 3, it is unclear what the exact meaning of each point is. Does each point represent an image in CIFAR-10? The authors should provide an explanation in the caption.
- In Appendix A.2, Fig. 6 should be referenced instead of Fig. 7. I cannot understand why PGD+SLIM is stronger than PGD based on Fig. 6, as many red dots are completely overlapped by blue dots in PGD. I recommend using smaller dots to avoid overlapping.

7. The considerable divergence in reported results from those in other studies, coupled with the ambiguity surrounding the computation of the Hessian's trace, raises concerns about the reproducibility of the findings.

[1] Moosavi-Dezfooli, Seyed-Mohsen, et al. "Robustness via curvature regularization, and vice versa." CVPR. 2019.
[2] Yao, Zhewei, et al. "Hessian-based analysis of large batch training and robustness to adversaries." NIPS. 2018.
[3] Miyato, Takeru, et al. "Virtual adversarial training: a regularization method for supervised and semi-supervised learning." IEEE transactions on pattern analysis and machine intelligence 41.8 (2018): 1979-1993.
[4] Addepalli, Sravanti, et al. "Towards achieving adversarial robustness by enforcing feature consistency across bit planes." CVPR. 2020.

**Questions:**

See the weakness section.